# Mild Copper-Catalyzed, l-Proline-Promoted Cross-Coupling of Methyl 3-Amino-1-benzothiophene-2-carboxylate

**DOI:** 10.3390/molecules26226822

**Published:** 2021-11-11

**Authors:** Vilija Kederienė, Indrė Jaglinskaitė, Paulina Voznikaitė, Jolanta Rousseau, Patrick Rollin, Algirdas Šačkus, Arnaud Tatibouët

**Affiliations:** 1Department of Organic Chemistry, Kaunas University of Technology, Radvilėnų pl. 19, LT-50254 Kaunas, Lithuania; indrejag@gmail.com (I.J.); paulina.voznikaite@ktu.lt (P.V.); algirdas.sackus@ktu.lt (A.Š.); 2Univ. Artois, CNRS, Centrale Lille, Univ. Lille, UMR 8181—UCCS—Unité de Catalyse et Chimie du Solide, F-62300 Lens, France; jolanta.rousseau@univ-artois.fr; 3Institut de Chimie Organique et Analytique (ICOA), Université d’Orléans et CNRS, UMR 7311, BP 6759, F-45067 Orléans, France; patrick.rollin@univ-orleans.fr

**Keywords:** benzo[*b*]thiophene, copper catalysis, l-proline, *N*-arylation, aryl iodides

## Abstract

Cu-catalyzed *N*-arylation is a useful tool for the chemical modification of aromatic heterocycles. Herein, an efficient carbon–nitrogen cross-coupling of methyl 3-amino-1-benzothiophene-2-carboxylate with a range of (hetero)aryl iodides using CuI, l-proline and Cs_2_CO_3_ in dioxane at moderate temperature is described. The procedure is an extremely general, relatively cheap, and experimentally simple way to afford the *N*-substituted products in moderate to high yields. The structures of the new heterocyclic compounds were confirmed by NMR spectroscopy and HRMS investigation.

## 1. Introduction

Compounds containing the benzothiophene frame are important building blocks in organic synthesis and are used in a broad range of applications [1]. The benzothiophene skeleton is present in many organic materials, such as solar cells, light-emitting diodes (OLEDs), and field-effect transistors [2,3,4]. In the medical field, synthetic benzothiophene-based derivatives show various biological activities [5]. Recently, they were evaluated as inhibitors of tubulin polymerization [6] and as candidates for the treatment of human cancer [6]; furthermore, they can be used for anti-HCV (hepatitis C virus) applications [7] as anti-inflammatory agents [8]. Moreover, a number of aminobenzothiophene derivatives are recognized as antioxidant **I** [9,10], photochemotherapeutic **II** [11], antimitotic **III** [12], antimicrobial **IV** [13], antiproliferative **V** [14], and anti-inflammatory **VI** agents [9,15] (Figure 1).

The importance of the aforementioned class of compounds supported our interest in the synthesis of the anthranilic acid analogues in which the thiophene moiety is present as a benzene ring bioisostere [16]. Moreover, *N*-substituted anthranilic acids are broadly used in the treatment of various diseases, such as arthritis, atypical dermatitis, and bronchial asthma, and they are also described as anti-allergic, analgesic, and anticancer agents [17,18,19,20,21].

Many methods have been developed for the synthesis of anthranilic acids. Copper-catalyzed arylation is one of the most powerful tools for the synthesis of *N*-arylanthranilic acids, which are important precursors for the synthesis of substituted acridines and acridones [22,23,24,25,26,27].

The traditional Ullmann-type reaction requires specific reaction conditions, such as strong base and high temperature, and offers a limited substrate scope [28]. Following important earlier breakthroughs, a major improvement came in 2001 with the discovery of new versatile and efficient copper/ligand systems. For example, Buchwald and co-workers reported that diamine ligands enable the formation of the C–N bond under much milder Cu-catalyzed conditions [29,30,31]. The same year, Venkataraman and co-workers discovered that the coupling of aryl halides with various nucleophiles could be successfully performed with 1,10-phenanthroline as ligand [32]. Ma and co-workers described that, by using l-proline as an additive, the copper-catalyzed cross-coupling reactions under mild conditions gave *N*-arylazoles in excellent yields [33,34]. This discovery was followed by several studies to improve C–N coupling conditions with the effect of l-proline as a ligand [35,36,37]. In view of these research studies, copper-catalyzed *N*-arylation of amines has become one of the most attractive and highly efficient methods to synthesize these compounds due to excellent catalytic selectivity and high functional group compatibility, as well as low toxicity and economic attractiveness [36].

To date, *N*-substituted benzo[*b*]thiophene derivatives have been synthesized using palladium-ligand complexes by the C–N coupling method of 3-halobenzo[*b*]thiophenes with anilines and aminopyridines [38,39,40,41,42] and the substitution of aminobenzo[*b*]thiophenes with bromobenzenes and bromopyridines [43,44]. Coupling reactions were successful; however, this synthetic pathway requires relatively expensive, air-sensitive, and highly toxic Pd catalysts systems [45]. Moreover, there exist difficulties with removing palladium traces during the purification stage, which is usually time-demanding and expensive; for this reason, this poses a serious practical problem in the pharmaceutical industry [46,47].

For the above reasons, we wish to report mild copper-catalyzed cross-coupling of aminobenzo[*b*]thiophene derivatives with (hetero)aryl iodides bearing various functional groups. The application of an Ullmann-type methodology to heteroaromatic systems is still relatively rare.

## 2. Results and Discussion

Here, we report an efficient catalytic system, which makes the methodology environmentally benign and economical, using commercially available copper(I) iodide and natural amino acid. Many copper complexes are effective catalyst precursors for *N*-arylation reactions. However, copper(I) salts give superior results, compared to copper(II), as they are a more active species [48]. We chose to focus on the use of CuI owing to its stability in air and lower strictness of the reaction conditions.

The precursor methyl 3-amino-1-benzothiophene-2-carboxylate **1** was synthesized under modified conditions by treatment of 2-fluorobenzonitrile with methyl thioglycolate in DMF [49,50].

The coupling reaction between methyl 3-amino-1-benzothiophene-2-carboxylate **1** and 4-iodoanisole was chosen as a model to optimize the reaction conditions with regard to bases, solvents, and ligands (Figure 1, Table 1). Several details are worth commenting on. Firstly, a strongly electron-donating agent, *N*,*N’*-dimethylethylenediamine (DMEDA), was initially investigated as an additive for the present reaction because of its known and excellent activity in CuI-catalyzed cross-coupling [51]. Since a sensitive carboxylate group is present in substrate **1** [49], we used milder standard bases (K_3_PO_4_, Cs_2_CO_3_, and K_2_CO_3_), which are often used for catalytic arylation. The coupling reaction proceed well in the presence of Cs_2_CO_3_ as a base in dioxane at a reflux temperature for 24 h; the expected *N*-arylated coupling product **3a** was obtained in 31% yield. Other bases commonly employed for these couplings, such as K_2_CO_3_ and K_3_PO_4_, were inferior to Cs_2_CO_3_, affording **3a** in 5 and 15% yield, respectively (Table 1, entries 1–3). Therefore, Cs_2_CO_3_ was selected as the base of choice for further screening reactions.

In the assessment of the effect of the solvent, the coupling results showed that dioxane was the most effective solvent, while DMSO and DMF gave only traces of the desired coupling product (Table 1, compare entries 3–6). On the other hand, dioxane is easily removable from reaction mixtures. Several bidentate *N*,*N*-ligands (DMEDA [51] and 10-phenanthroline [52]), which are commonly used as standard ligands in Cu-catalyzed cross-coupling reactions, were examined and significant differences in yield were observed (Table 1, entries 3 and 7). Conversely, amino acids can also act as effective bidentate N,O-chelators in the C–N coupling [53]; therefore, we focused our attention on employing l-proline as the additive. As expected, when N,O-donor l-proline was used, the reaction provided the coupling product in a much higher yield, 58% (Table 1, entry 8). l-proline capability for promoting Cu-catalyzed C–N coupling reactions might be dependent on its reactivity as the coupling agent and coordination ability as a bidentate additive [53]. In the literature has been reported an accelerating effect induced by the structure of amino acids and possible catalytic mechanisms for Cu-catalyzed amino acids promoted coupling reactions [33,34,53].

To compare the reactivity, we also studied the copper-catalyzed amination using methyl 3-iodo-1-benzothiophene-2-carboxylate **2** and 4-anisidine as coupling reagent (Table 1, entry 9). The precursor **2** was already known [54] and was synthesized, here, in a higher yield, 73%, via a diazotization–iodination reaction of compound **1** (Figure 1). However, to our surprise, cross-coupling was not successful, as compound **2** afforded only 12% of product **3a**, which, we suppose, was possibly because of a radical reaction mechanism [55]. The literature has described several speculative mechanisms for Ullmann-type coupling reactions for the aryl halide activation step and one of them is a possible radical pathway-atom transfer mechanism involving aryl radical, which causes the transfer of the halide atom from the aryl halide [56]. Methyl 3-iodo-1-benzothiophene-2-carboxylate **2** may form an inactive radical, which leads to the formation of **3a** in a very low yield.

The coupling reaction of methyl 3-amino-1-benzothiophene-2-carboxylate (**1**) with 4-bromoanisole was also tested. Under these conditions, 4-bromoanisole was less reactive, affording coupled product **3a** in a 38% yield (Table 1, entry 10), while 4-iodoanisole exhibited greater reactivity. These results established the reactivity order of the aryl halides: iodides > bromides > chlorides [57].

It also seems that the reaction time played an important role. After 6 h of heating using 4-iodoanisole, the yield of the coupled product **3a** was only 28%, while, upon allowing the reaction to continue for 24 h, the yield was increased to 58% (Table 1, entries 8, 11–12). A longer reaction time (48 h) did not improve the efficiency of the coupling transformation.

For a copper-catalyzed *N*-arylation, the choice of a suitable stoichiometric quantity of copper catalyst and ligand is very important. For further optimization of the reaction conditions, the increase in the yields of the coupling products was observed by changing the molar ratio of CuI:l-proline (Figure 2). As summarized in Table 2, it was found that the ratio of 1:1 for CuI and l-proline was necessary to ensure a satisfactory result. It was noticed that, when different ratios of reagents were used, lower yields of reaction products were obtained (Table 2, compare entries 2–5). Moreover, the ratio of Cu(I) to l-proline used in this study was only equal to 1:1, whereas the ratio generally used in most of the other copper-catalyzed cross-coupling reactions using amino acids as ligands is 1:2 [53]. When 20 mol% of CuI and l-proline each was used, the reaction was complete in 24 h and gave methyl 3-[(4-methoxyphenyl)amino]-1-benzothiophene-2-carboxylate **3a** in excellent yield (Table 2, entry 2). Reducing the amounts of copper catalyst and l-proline as a promoter to 10 mol% gave a similar yield (Table 2, entry 6), while 5 mol% afforded slightly worse results (Table 2, entry 7). Thus, the appropriate ratio of CuI:l-proline is essential for the optimal reaction outcome. More importantly, we were pleased to observe that the control experiment with methyl 3-amino-1-benzothiophene-2-carboxylate **1** without ligand was also successful (Table 2, entry 1).

To improve the reaction yield, as well as to shorten the reaction time, an attempt was made to use microwave (MW) irradiation heating (Figure 3, Table 3). Our previous results showed that dioxane and acetonitrile were the more suitable solvents, while THF, toluene, or DMF gave poor yields of coupling products under MW heating. According to the literature, dioxane is microwave-transparent, and the reaction in this solvent can be thermally activated only if other components in the reaction mixture respond to microwave energy, i.e., if the reaction mixture contains either polar reactants or ions [58]. However, in our case, despite the presence of polar amine and Cs_2_CO_3_ in the reaction mixture, MW activation was not very effective. MW heating using acetonitrile did not increase the yields of the coupling products either, contrary to using conventional heating (Table 3). For this reason, further experiments were carried out using conventional heating in dioxane for 24 h.

Thus, the optimized reaction conditions utilized 10 mol% of Cu(I), 10 mol% of l-proline, and Cs_2_CO_3_ (2 equiv) in dioxane at reflux under argon. These reaction conditions were applied to the coupling of methyl 3-amino-1-benzothiophene-2-carboxylate **1** with various functionalized aryl iodides to synthesized coupling compound **3a**–**s** (Figure 2, Table 3 and Table 4). To study the steric and electronic effects on the product selectivity, reactions with aryl iodides having electron-donating and electron-withdrawing groups such as NO_2_, OCH_3_, CF_3_, F, OH, and NH_2_ were further examined. Table 3 and Table 4 show that all *para*-, *meta*-, and *ortho*-substitutions on the aryl halide were tolerated under these reaction conditions. For example, the palladium-catalyzed C–N coupling of *ortho*-substituted aryl halides is difficult and is the major drawback of this method [59]. We showed that *N*-substituted methyl 3-amino-1-benzothiophene-2-carboxylates **3a**–**o** can be obtained in moderate to high yields using activated aryl iodides as starting reagents.

The steric hindrance did not seem to influence cross-coupling, as the reaction of aminobenzothiophene **1** with *ortho*-, *meta*-, or *para*-substituted iodides gave products in similar yields. It is noteworthy that activated iodides bearing electron-withdrawing groups such as iodoanisoles gave good yields (Table 3, entries 1–4, and Table 4, entry 1). The coupling of non-substituted phenyl iodide with compound **1** gave a lower yield of the desired product **3l** (Table 4, entry 6).

A variety of functional groups can be present on aryl halides; nonetheless, for electron-rich aryl iodides bearing free hydroxy and amino functional groups, low conversion was observed under the above-described standard reaction conditions (Table 4, entries 7 and 9). After some experimentation, we found out that this problem could be easily solved by the protection of those functional groups (Table 4, entries 8 and 10). For that purpose, *N*-(4-iodophenyl)acetamide and 4-iodophenyl acetate were prepared from 4-iodophenol and 4-iodoaniline, respectively, following known procedures [60]. Acylation of the –OH and –NH_2_ groups proved to be crucial and resulted in the formation of the products **3m** and **3o** in higher yield (Table 4, entries 8, 10). The cross-coupling approach of compound **1** with 4-iodophenyl acetate formed the same final product **3m** as coupling with 2-iodophenol (Table 4, entries 7–8). Coupling reactions of methyl 3-amino-1-benzothiophene-2-carboxylate **1** with 2-iodoaniline, *N*-(2-iodophenyl)acetamide and 2-iodophenol were unsuccessful and gave traces of the products (Table 4, entries 11–13).

However, the Ullmann-type reaction between 2-iodoaniline and compound **1** after additional heating for 72 h gave an interesting tetracyclic derivative **4** in a 41% yield (Figure 4). This new diazepinone **4** was the result of a C–N coupling, followed by spontaneous intramolecular condensation in one step, with loss of methanol. Careful NMR analysis permitted the elucidation of structure **4**. The combined application of HSQC, HMBC, and COSY experiments finally enabled the unambiguous assignment of 5a,7-dihydro-6*H*-benzothieno[3,2-*b*][1,5]benzodiazepin-6-one **4**.

The study of the literature revealed only a few recently described examples via the C–N cross-coupling of diazepinone compounds. For example, recently reported synthetic pathways to dibenzodiazepine analogues required catalytic quantities of palladium [61]. Typically, several steps are required to prepare dibenzodiazepinones, beginning with the copper-catalyzed coupling reactions of anthranilic acids or substituted benzoates [62]. On the other hand, double amination of *ortho*-substituted aryl bromides has been reported, which revealed a strong *ortho*-substituent effect caused by *N*-aryl aminocarbonyl groups during copper-catalyzed aryl amination [63]. To our knowledge, only one example of a tricyclic benzothieno[3,2-*b*]diazepinone derivative has been described and was studied as an antiherpetic agent [64]. For this reason, we are currently working on the improved synthesis of benzothieno[3,2-*b*][1,5]benzodiazepinones via copper-catalyzed cross-coupling, whose results will be reported in due course.

The reaction of methyl 3-amino-1-benzothiophene-2-carboxylate **1** with 2-iodobenzoic acid using the above conditions failed to give any conversion to the desired product. The possible explanation could be that coordination of the resulting aryl carboxylate to copper resulted in the inactivation of the catalytic species and that the aryl carboxylate might have limited solubility in dioxane [51].

The optimized Cu-catalyzed reaction conditions also were applied to the coupling of methyl 3-amino-1-benzothiophene-2-carboxylate **1** with hetero iodide **2** to synthesized coupling compound **5** (Figure 5).

The result indicated that methyl 3-amino-1-benzothiophene-2-carboxylates **1** must play an important role in accelerating the Cu(I)-catalyzed cross-coupling reactions. This effect might be influenced by a possible chelating effect of the aminobenzo[*b*]thiophene N atom, the Cu complex, and the oxygen atom of the 2-carbonyl unit. For example, Liebeskind and co-workers reported Ullmann-type cross-coupling reactions of substituted aryl, heteroaryl, and alkenyl halides using copper(I)-thiophene-2-carboxylate (CuTC) as a promoter [65]. According to the literature, Liebeskind’s catalyst CuTC is superior to other catalysts in various cross-couplings [66].

## 3. Materials and Methods

### 3.1. General Information

Reactions in anhydrous conditions were performed under argon atmosphere in pre-dried flasks, using anhydrous solvents (distilled when necessary according to D. D. Perrin, W. L. F. Armarego and D. R. Perrin in *Purification of Laboratory Chemicals*, Pergamon, Oxford, 1986). All reagents and solvents were purchased from commercial chemical suppliers and used without further purification. The course of the reactions was monitored by initial TLC analysis on aluminum foil-backed plates (Merck Kieselgel 60 F_254_, Darmstadt, Germany), which were developed using standard visualization techniques or agents: UV fluorescence (254 nm) and by staining with a 1% aq KMnO_4_ solution. Flash column chromatography was performed with Silica Gel 60 Å (230–400 µm, Merck KGaA, Darmstadt, Germany). Melting points [°C] were measured with a Thermo Scientific 9200 capillary apparatus, Büchi Melting Point M-560 (Büchi Labortechnik AG, Flawil, Switzerland) (for compounds **3c**, **3e**, **3f**), and are uncorrected. NMR spectra were recorded on a 400 MHz Bruker Avance 2 400 and Bruker Avance III 400 (for compounds **3c**, **3e**, **3f**) spectrometers (Bruker BioSpin AG, Fallanden, Switzerland) (400 MHz (^1^H), 100 MHz (^13^C), 376 MHz (^19^F)). Chemical shifts are expressed in parts per million (ppm) downfield from TMS internal standard. Coupling patterns for ^1^H-NMR are designated as s = singlet, d = doublet, t = triplet, m = multiplet, (some ^13^C-NMR are designated as d = doublet, q = quartet), and coupling constants are given in Hz. NMR peak assignments were elucidated via DEPT, COSY, and HSQC techniques when necessary. IR spectra from samples in neat form were measured with a Thermo Scientific Nicolet iS10 FT-IR spectrophotometer (Thermo Fisher Scientific Inc., Waltham, MA, USA) and Bruker Tensor 27 (Bruker Optic GmbH, Ettlingen, Germany) (KBr pellets, for compounds **3c**, **3e**, **3f**). IR absorption frequencies are given in cm^−1^. Low-resolution mass spectra were recorded with Perkin–Elmer Sciex API 300 spectrometer (PerkinElmer Inc., Waltham, MA, USA) and Shimadzu LCMS-2020 (Shimadzu Corporation, Kyoto, Japan) (for compounds **3c**, **3e**, **3f**). High-resolution mass spectra (HRMS) were recorded with a Bruker MaXis spectrometer (Bruker Daltonik GmbH, Bremen, Germany) in the electrospray ionization (ESI) mode. ^1^H-, ^13^C-, ^19^F-, DEPT, and some COSY, HSQC NMR spectra, and HRMS data of all new compounds are provided in Appendix A.

### 3.2. Synthesis of Methyl 3-Amino-1-benzothiophene-2-carboxylate *(**1**)*

To a solution of 2-fluorobenzonitrile (0.45 mL, 4.15 mmol) in dry DMF (2 mL) at room temperature methyl thioglycolate (0.41 mL, 4.58 mmol) was added dropwise. The reaction mixture was stirred at room temperature for 10 min, and K_2_CO_3_ (680 mg, 4.92 mmol) was added. After stirring at 105 °C for 2 h, the reaction mixture was poured into crushed ice-water (50 mL). The resulting precipitate was collected by filtration, rinsed with cold water, and air dried to give compound **1** as a pale beige solid; yield: 840 mg, 98%; m.p. 107–108 °C (lit. [67] m.p. 110–111 °C). MS (ESI): *m*/*z* = 208 [M + H]^+^.

### 3.3. Synthesis of Methyl 3-Iodo-1-benzothiophene-2-carboxylate *(**2**)*

Methyl 3-amino-1-benzothiophene-2-carboxylate (601 mg, 2.90 mmol) was gradually added to a vigorously stirred 6 M hydrochloric acid solution (2 mL). The reaction mixture was stirred for 20 min at 60 °C. Then mixture was cooled below 0 °C (ice and NaCl bath), and slowly diazotized with sodium nitrite (219 mg, 3.186 mmol). The resulting diazonium salt was stirred for 1 h at this temperature and was poured at once into a well stirred solution of potassium iodide (721 mg, 4.343 mmol) in water (2 mL). The reaction mixture was heated at 60 °C and once evolution of nitrogen had finished, resulting reaction mixture was cooled. Then, mixture extracted with chloroform (3 × 30 mL) and washed with water (50 mL), brine, and finally dried over MgSO_4_. After filtration, the solvent was removed under reduced pressure. The obtained residue was purified by column chromatography, *R_f_* = 0.24 (PE/EtOAc 97:3). Yield: 700 mg, 73%; pale yellow solid; m.p. 92–93 °C (lit. [54] m.p. 92–93.5 °C). ^1^H-NMR (400 MHz, CDCl_3_): *δ* = 3.98 (s, 3H, OCH_3_), 7.47–7.54 (m, 2H, 5-H, 6-H), 7.80–7.83 (m, 1H, 7-H), 7.96–7.98 (m, 1H, 4-H) ppm. ^13^C-NMR (100 MHz, CDCl_3_): *δ* = 52.8 (OCH_3_), 88.4 (C_q_), 122.8 (C-4), 126.1 (C-6), 128.3 (C-5, C-7), 130.2 (C_q_), 140.4 (C_q_), 142.0 (C_q_), 162.2 (C=O) ppm. IR (neat): *ν_max_* = 1713 (C=O), 1497, 1218 (C-O), 1082, 1052 (C=C) cm^−1^. MS (ESI): *m*/*z* = 319 [M + H]^+^. HRMS (ESI): *m*/*z* [M + H]^+^ calcd. for C_10_H_8_IO_2_S 318.92842, found 318.92847. HRMS (ESI): *m*/*z* [M + Na]^+^ calcd. for C_10_H_7_INaO_2_S 340.91036, found 340.91040. Compound **2** was previously prepared using a different synthetic pathway. The ^1^H- and ^13^C-NMR spectra are in good agreement with the literature data [54].

### 3.4. Synthesis Procedure for the Preparation of Compounds ***3a**–**o***

Under argon atmosphere, a mixture of methyl 3-amino-1-benzothiophene-2-carboxylate (1 equiv), aryl iodide (1.5 equiv), Cs_2_CO_3_ (2 equiv), CuI (0.1 equiv), and l-proline (0.1 equiv) in dry dioxane (2 mL) was stirred under reflux for 24 h. After the mixture was cooled to room temperature and diluted with water and extracted with ethyl acetate. The combined organic layer was washed with brine, dried over anhydrous Na_2_SO_4_, and the solvent was evaporated under reduced pressure. The resulting residue was purified by column chromatography using PE/EtOAc as eluent to afford the corresponding compounds. The data for selected compounds is described below.

#### 3.4.1. Methyl 3-[(4-Methoxyphenyl)amino]-1-benzothiophene-2-carboxylate (**3a**)

The representative experimental procedure was applied to compound **1** (200 mg, 0.965 mmol) to yield **3a** (260 mg, 86%); *R_f_* = 0.30 (PE/EtOAc 97:3); yellow solid, m.p. 103–104 °C. ^1^H-NMR (400 MHz, CDCl_3_): *δ* = 3.83 (s, 3H, OCH_3_), 3.91 (s, 3H, COOCH_3_), 6.86 (d, *J* = 8.9 Hz, 2H, 2-H_Ph_, 6-H_Ph_), 7.06–7.10 (m, 3H, 3-H_Ph_, 5-H_Ph_, 6-H), 7.20 (d, *J* = 8.2 Hz, 1H, 7-H), 7.35–7.39 (m, 1H, 5-H), 7.72 (d, *J* = 8.2 Hz, 1H, 4-H), 8.79 (s, 1H, NH) ppm. ^13^C-NMR (100 MHz, CDCl_3_): *δ* = 51.9 (COO*C*H_3_), 55.7 (OCH_3_), 103.5 (C_q_), 114.5 (C_Ph_-3, C_Ph_-5), 123.4 (C-4, C-7), 125.2 (C_Ph_-2, C_Ph_-6), 125.8 (C-6), 127.7 (C-5), 131.7 (C_q_), 135.1 (C_q_), 140.5 (C_q_), 148.0 (C_q_), 157.0 (C_q_), 166.2 (C=O) ppm. IR (neat): *ν_max_* = 3297 (N-H), 1666 (C=O), 1577, 1508, 1440, 1397, 1230 (C–O), 1146, 1031 (C=C, C–N) cm^−1^. MS (IS): *m*/*z* = 282 [M–OCH_3_]^+^, 314 [M + H]^+^, 336 [M + Na]^+^. HRMS (ESI): *m*/*z* [M + H]^+^ calcd. for C_17_H_16_NO_3_S 314.08454, found 314.08440. HRMS (ESI): *m*/*z* [M + Na]^+^ calcd. for C_17_H_15_NNaO_3_S 336.06649, found 336.06626.H]^−^, 95%). HRMS (ESI^+^) for C_21_H_29_N_3_NaO_5_ ([M + Na]^+^) calcd 426.1999, found 426.2001.

#### 3.4.2. Methyl 3-[(2-Methoxyphenyl)amino]-1-benzothiophene-2-carboxylate (**3b**)

The representative experimental procedure was applied to compound **1** (200 mg, 0.965 mmol) to yield **3b** (218 mg, 72%); *R_f_* = 0.22 (PE/EtOAc 98:2); yellow solid; m.p. 123–124 °C. ^1^H-NMR (400 MHz, CDCl_3_): *δ* = 3.89 (s, 3H, COOCH_3_), 3.91 (s, 3H, OCH_3_), 6.76–6.80 (m, 1H, H_Ph_), 6.88 (dd, *J* = 0.9 Hz, *J* = 7.8 Hz, 1H, 6-H_Ph_), 6.93 (dd, *J* = 1.1 Hz, *J* = 8.0 Hz, 1H, 3-H_Ph_), 6.97–7.02 (m, 1H, H_Ph_), 7.14–7.18 (m, 1H, 6-H), 7.40 (td, *J* = 0.95 Hz, *J* = 8.1 Hz, 1H, 5-H), 7.52 (d, *J* = 8.3 Hz, 1H, 7-H), 7.74 (d, *J* = 8.1 Hz, 1H, 4-H), 8.56 (s, 1H, NH) ppm. ^13^C-NMR (100 MHz, CDCl_3_): *δ* = 52.1 (COO*C*H_3_), 55.9 (OCH_3_), 108.0 (C_q_), 111.0 (C_Ph_-3), 119.7 (C_Ph_-6), 120.5 (C_Ph_), 123.0 (C-4), 123.4 (C_Ph_), 123.6 (C-6), 125.9 (C-7), 127.8 (C-5), 131.8 (C_q_), 132.6 (C_q_), 140.1 (C_q_), 145.6 (C_q_), 150.7 (C_q_), 165.6 (C=O) ppm. IR (neat): *ν_max_* = 3378 (N–H), 1684 (C=O), 1563, 1530, 1502, 1437, 1274, 1239 (C-O), 1218, 1109, 1025 (C=C, C–N) cm^−1^. MS (ESI): *m*/*z* = 282 [M–OCH_3_]^+^, 314 [M + H]^+^, 336 [M + Na]^+^, 352 [M + K]^+^. HRMS (ESI): *m*/*z* [M + H]^+^ calcd. for C_17_H_16_NO_3_S 314.08454, found 314.08405.

#### 3.4.3. Methyl 3-[(4-Flourophenyl)amino]-1-benzothiophene-2-carboxylate (**3c**)

The representative experimental procedure was applied to compound **1** (200 mg, 0.965 mmol) to yield **3c** (273 mg, 94%); *R_f_* = 0.22 (PE/EtOAc 98:2); yellowish crystals; m.p. 92–93 °C. ^1^H-NMR (400 MHz, CDCl_3_): *δ* = 3.95 (s, 3H, OCH_3_), 6.99–7.05 (m, 2H, H_Ar_), 7.06–7.11 (m, 2H, H_Ar_), 7.14–7.18 (m, 1H, H_BT_), 7.26–7.32 (m, 1H, H_BT_), 7.43–7.49 (m, 1H, H_BT_), 7.77 (d, *J* = 8.2 Hz, 1H, 4-H_BT_), 8.75 (s, 1H, NH) ppm. ^13^C-NMR (100 MHz, CDCl_3_): *δ* = 52.0 (OCH_3_), 105.9 (C_q_), 115.9 (d, ^2^*J*_C,F_ = 23.0 Hz, 2xCH), 123.5 (d, ^3^*J*_C,F_ = 14.0 Hz, CH), 124.1 (d, ^3^*J*_C,F_ = 8.0 Hz, CH), 125.6 (CH), 127.8 (CH), 131.7 (C_q_), 138.4 (d, ^3^*J*_C,F_ = 2.0 Hz, C_q_), 140.3 (C_q_), 146.8 (C_q_), 159.7 (d, ^1^*J*_C,F_ = 242.0 Hz, C_q_), 166.1 (C=O) ppm. ^19^F-NMR (376 MHz, CDCl_3_): *δ* = −119.03 (s, 1F) ppm. IR (KBr): *ν_max_* = 3313 (N–H), 1674 (C=O), 1674, 1433, 1278, 1253, 1242, 1147, (C–O, C–F, C=C, C–N) cm^−1^. MS (ESI): *m*/*z* = 302 [M + H]^+^.

#### 3.4.4. Methyl 3-[(2-Fluorophenyl)amino]-1-benzothiophene-2-carboxylate (**3d**)

The representative experimental procedure was applied to compound **1** (200 mg, 0.965 mmol) to yield **3d** (230 mg, 79%); *R_f_* = 0.45 (PE/EtOAc 98:2); yellowish crystals; m.p. 126–127 °C. ^1^H-NMR (400 MHz, CDCl_3_): *δ* = 3.93 (s, 3H, OCH_3_), 6.96–7.06 (m, 3H, 4-H_Ph_, 5-H_Ph_, 6-H_Ph_), 7.12–7.21 (m, 2H, 3-H_Ph_, 6-H), 7.41–7.45 (m, 2H, 5-H, 7-H), 7.78 (d, *J* = 8.3 Hz, 1H, 4-H), 8.53 (s, 1H, NH) ppm. ^13^C-NMR (100 MHz, CDCl_3_): *δ* = 52.1 (OCH_3_), 108.4 (C_q_), 116.0 (d, ^2^*J*_C,F_ = 19.1 Hz, C_Ph_-3), 122.2 (d, ^4^*J*_C,F_ = 1.4 Hz, C_Ph_-5), 123.4 (C-4), 123.8 (C-6), 123.8 (d, ^3^*J*_C,F_ = 7.2 Hz, C_Ph_-4), 124.1 (d, ^3^*J*_C,F_ = 3.8 Hz, C_Ph_-6), 125.2 (C-7), 127.8 (C-5), 130.6 (d, ^2^*J*_C,F_ = 11.3 Hz, C_Ph_-1), 132.2 (C_q_), 140.0 (C_q_), 145.0 (C_q_), 155.0 (d, ^1^*J*_C,F_ = 245.5 Hz, C_Ph_-2), 165.6 (C=O) ppm. ^19^F-NMR (376 MHz, CDCl_3_): *δ* = −127.42 (s, 1F) ppm. IR (neat): *ν_max_* = 3324 (N–H), 1675 (C=O), 1433, 1277, 1243 (C–O), 1190 (C–F, C=C, C–N) cm^−1^. MS (ESI): *m*/*z* = 270 [M–OCH_3_]^+^, 302 [M + H]^+^, 324 [M + Na]^+^, 340 [M + K]^+^. HRMS (ESI): *m*/*z* [M + H]^+^ calcd. for C_16_H_13_FNO_2_S 302.06455, found 302.06373.

#### 3.4.5. Methyl 3-{[4-(Trifluoromethyl)phenyl]amino}-1-benzothiophene-2-carboxylate (**3e**)

The representative experimental procedure was applied to compound **1** (200 mg, 0.965 mmol) to yield **3e** (297 mg, 97%); *R_f_* = 0.43 (PE/EtOAc 98:2); yellowish crystals; m.p. 125–126 °C. ^1^H-NMR (400 MHz, CDCl_3_): *δ* = 3.96 (s, 3H, OCH_3_), 7.08 (d, *J* = 8.4 Hz, 2H, 2xH_Ar_), 7.28–7.31 (m, 1H, H_Ar_), 7.48–7.54 (m, 4H, H_Ar_, H_BT_), 7.82–7.85 (m, 1H, H_BT_), 8.68 (s, 1H, NH) ppm. ^13^C-NMR (100 MHz, CDCl_3_): *δ* = 52.3 (OCH_3_), 110.7 (C_q_), 119.3 (2 × CH), 123.5 (CH), 124.0 (CH), 124.3 (q, ^2^*J*_C,F_ = 33.0 Hz, C_q_), 125.4 (CH), 126.5 (q, ^3^*J*_C,F_ = 4.0 Hz, 2 × CH), 128.0 (CH), 132.2 (C_q_), 140.0 (C_q_), 143.9 (C_q_), 145.8 (C_q_), 165.5 (C=O) ppm. ^19^F-NMR (376 MHz, CDCl_3_): *δ* = −61.71 (s, 3F) ppm. IR (KBr): *ν_max_* = 3317 (N–H), 1668 (C=O), 1328, 1280, 1260, 1244, 1117, 1110 (C–F, C–O–C, C=C, C–N) cm^−1^. MS (ESI): *m*/*z* = 352 [M + H]^+^.

#### 3.4.6. Methyl 3-{[3-(Trifluoromethyl)phenyl]amino}-1-benzothiophene-2-carboxylate (**3f**)

The representative experimental procedure was applied to compound **1** (200 mg, 0.965 mmol) to yield **3f** (248 mg, 73%); *R_f_* = 0.45 (PE/EtOAc 98:2); yellowish crystals; m.p. 133–134 °C. ^1^H-NMR (400 MHz, CDCl_3_): *δ* = 3.93 (s, 3H, OCH_3_), 7.16–7.23 (m, 2H, H_Ar_), 7.28–7.31 (m, 2H, H_Ar_), 7.37 (dd, *J* = 8.0 Hz, *J* = 11.5 Hz, 2H_BT_), 7.43–7.49 (m, 1H, H_BT_), 7.80 (d, *J* = 8.1 Hz, 1H_BT_), 8.72 (s, 1H, NH) ppm. ^13^C-NMR (100 MHz, CDCl_3_): *δ* = 52.3 (OCH_3_), 109.4 (C_q_), 117.2 (q, ^3^*J*_C–F_ = 4.0 Hz, CH), 119.4 (q, ^3^*J*_C–F_ = 4.0 Hz, CH), 123.5 (CH), 123.6 (CH), 123.9 (CH), 124.0 (q, ^1^*J*_C–F_ = 272.0 Hz, C_q_), 125.3 (CH), 128.0 (CH), 129.7 (CH), 131.7 (q, ^2^*J*_C–F_ = 32.0 Hz, C_q_), 131.9 (C_q_), 140.1 (C_q_), 143.2 (C_q_), 144.6 (C_q_), 165.7 (C=O) ppm. ^19^F-NMR (376 MHz, CDCl_3_): *δ* = −62.86 (s, 3F) ppm. IR (KBr): *ν_max_* = 3312 (N–H), 1680 (C=O), 1337, 1275, 1165, 1143, 1120 (C–F, C–O–C, C=C, C–N) cm^−1^. MS (ESI): *m*/*z* = 352 [M + H]^+^.

#### 3.4.7. Methyl 3-{[2-(Trifluoromethyl)phenyl]amino}-1-benzothiophene-2-carboxylate (**3g**)

The representative experimental procedure was applied to compound **1** (200 mg, 0.965 mmol) to yield **3g** (224 mg, 72%); *R_f_* = 0.45 (PE/EtOAc 98:2); pale pink needles; m.p. 152–153 °C. ^1^H-NMR (400 MHz, CDCl_3_): *δ* = 3.92 (s, 3H, OCH_3_), 6.94 (d, *J* = 8.1 Hz, 1H, 6-H_Ph_), 7.11 (t, *J* = 7.6 Hz, 1H, 4-H_Ph_), 7.14–7.18 (m, 1H, 5-H_Ph_), 7.27–7.33 (m, 2H, 6-H, 7-H), 7.41 (td, *J* = 1.1 Hz, *J* = 8.2 Hz, 1H, 5-H), 7.67 (d, *J* = 7.8 Hz, 1H, 3-H_Ph_), 7.77 (d, *J* = 8.2 Hz, 1H, 4-H), 8.82 (s, 1H, NH) ppm. ^13^C-NMR (100 MHz, CDCl_3_): *δ* = 52.4 (OCH_3_), 110.4 (C_q_), 121.3 (q, ^2^*J*_C,F_ = 29.8 Hz, C_Ph_-2), 122.3 (C_Ph_-6), 122.7 (C_Ph_-4), 123.6 (C-4), 124.0 (C_Ph_-5), 124.5 (q, ^1^*J*_C,F_ = 272.8 Hz, CF_3_), 125.4 (C-7), 126.9 (q, ^3^*J*_C,F_ = 5.3 Hz, C_Ph_-3), 127.9 (C-5), 132.1 (C_q_), 132.5 (C-6), 140.1 (C_q_), 141.0 (C_q_), 144.2 (C_q_), 165.6 (C=O) ppm. ^19^F-NMR (376 MHz, CDCl_3_): *δ* = −62.01 (s, 3F) ppm. IR (neat): *ν_max_* = 3316 (N–H), 1678 (C=O), 1588, 1463, 1321, 1293, 1274, 1251 (C–O), 1115 (C–F), 1102, 1033 (C=C, C–N) cm^−1^. MS (ESI): *m*/*z* = 320 [M–OCH_3_]^+^, 352 [M + H]^+^, 374 [M + Na]^+^, 390 [M + K]^+^. HRMS (ESI): *m*/*z* [M + H]^+^ calcd. for C_17_H_13_F_3_NO_2_S 352.06136, found 352.06111. HRMS (ESI): *m*/*z* [M + Na]^+^ calcd. for C_17_H_12_F_3_NNaO_2_S 374.04331, found 374.04331. HRMS (ESI): *m*/*z* [M + K]^+^ calcd. for C_17_H_12_F_3_KNO_2_S 390.01724, found 390.01729.

#### 3.4.8. Methyl 3-[(3-Methoxyphenyl)amino]-1-benzothiophene-2-carboxylate (**3h**)

The representative experimental procedure was applied to compound **1** (200 mg, 0.965 mmol) to yield **3h** (230 mg, 76%); *R_f_* = 0.25 (PE/EtOAc 98:2); pale beige solid; m.p. 89–90 °C. ^1^H-NMR (400 MHz, CDCl_3_): *δ* = 3.74 (s, 3H, OCH_3_), 3.92 (s, 3H, COOCH_3_), 6.62–6.66 (m, 3H, H_Ph_), 7.15–7.20 (m, 2H, H_Ph_, 6-H), 7.39–7.43 (m, 1H, 5-H), 7.47 (d, *J* = 8.3 Hz, 1H, 7-H), 7.75 (d, *J* = 8.2 Hz, 1H, 4-H), 8.71 (s, 1H, NH) ppm. ^13^C-NMR (100 MHz, CDCl_3_): *δ* = 52.0 (COO*C*H_3_), 55.4 (OCH_3_), 107.1 (C_q_), 107.2 (C_Ph_), 109.3 (C_Ph_), 114.1 (C_Ph_), 123.3 (C_Ph_), 123.6 (C-4), 125.9 (C-7), 127.8 (C-5), 129.9 (C-6), 132.2 (C_q_), 140.1 (C_q_), 143.7 (C_q_), 146.1 (C_q_), 160.5 (C_q_), 165.9 (C=O) ppm. IR (neat): *ν_max_* = 3310 (N–H), 1669 (C=O), 1574, 1536, 1435, 1391, 1274, 1238 (C–O), 1196, 1036 (C=C, C–N) cm^−1^. MS (ESI): *m*/*z* = 282 [M–OCH_3_]^+^, 314 [M + H]^+^. HRMS (ESI): *m*/*z* [M + H]^+^ calcd. for C_17_H_16_NO_3_S 314.08454, found 314.08484. HRMS (ESI): *m*/*z* [M + Na]^+^ calcd. for C_17_H_15_NNaO_3_S 336.06649, found 336.06691.

#### 3.4.9. Methyl 3-[(4-Nitrophenyl)amino]-1-benzothiophene-2-carboxylate (**3i**)

The representative experimental procedure was applied to compound **1** (200 mg, 0.965 mmol) to yield **3i** (190 mg, 60%); *R_f_* = 0.20 (PE/EtOAc 9:1); yellow solid; m.p. 139–140 °C (lit. [43] 133–135 °C). ^1^H-NMR (400 MHz, CDCl_3_): *δ* = 3.94 (s, 3H, OCH_3_), 6.98 (d, *J* = 8.9 Hz, 2H, 2-H_Ph_, 6-H_Ph_), 7.31 (t, *J* = 7.6 Hz, 1H, 6-H), 7.49–7.52 (m, 2H, 5-H, 7-H), 7.84 (d, *J* = 8.3 Hz, 1H, 4-H), 8.14 (d, *J* = 8.9 Hz, 2H, 3-H_Ph_, 5-H_Ph_), 8.65 (s, 1H, NH) ppm. ^13^C-NMR (100 MHz, CDCl_3_): *δ* = 52.5 (OCH_3_), 113.6 (C_q_), 117.7 (C_Ph_-2, C_Ph_-6), 123.7 (C-4), 124.5 (C-6), 125.1 (C-5), 125.7 (C_Ph_-3, C_Ph_-5), 128.2 (C-7), 132.3 (C_q_), 139.8 (C_q_), 141.9 (C_q_), 142.0 (C_q_), 148.9 (C_q_), 165.1 (C=O) ppm. IR (neat): *ν_max_* = 3340 (N-H), 1686 (C=O), 1589 (N–O), 1507, 1318 (N–O), 1238 (C–O), 1111 (C=C, C–N) cm^−1^. MS (ESI): *m*/*z* = 329 [M + H]^+^. HRMS (ESI): *m*/*z* [M + H]^+^ calcd. for C_16_H_13_N_2_O_4_S 329.05905, found 329.05911. HRMS (ESI): *m*/*z* [M + Na]^+^ calcd. for C_16_H_12_N_2_NaO_4_S 351.04100, found 351.04112. ^1^H- and ^13^C-NMR spectra are in good agreement with the literature data [43].

#### 3.4.10. Methyl 3-[(3-Nitrophenyl)amino]-1-benzothiophene-2-carboxylate (**3j**)

The representative experimental procedure was applied to compound **1** (200 mg, 0.965 mmol) to yield **3j** (241 mg, 79%); *R_f_* = 0.33 (PE/EtOAc 9:1); yellow solid; m.p. 141–142 °C. ^1^H-NMR (400 MHz, CDCl_3_): *δ* = 3.94 (s, 3H, OCH_3_), 7.23 (dd, *J* = 1.0 Hz, *J* = 8.2 Hz, 1H, 6-H_Ph_), 7.29 (dd, *J* = 1.4 Hz, *J* = 8.1 Hz, 1H, 4-H_Ph_), 7.39–7.43 (m, 2H, 5-H, H_Ph_), 7.47 (td, *J* = 1.0 Hz, *J* = 8.1 Hz, 1H, 6-H), 7.81–7.83 (m, 2H, 7-H, H_Ph_), 7.87 (dd, *J* = 1.0 Hz, *J* = 8.1 Hz, 1H, 4-H), 8.70 (s, 1H, NH) ppm. ^13^C-NMR (100 MHz, CDCl_3_): *δ* = 52.3 (OCH_3_), 111.0 (C_q_), 114.4 (C-7), 117.2 (C-4), 123.7 (C_Ph_), 124.2 (C_Ph_-6), 125.0 (C_Ph_), 125.5 (C_Ph_-4), 128.1 (C-6), 129.9 (C-5), 131.9 (C_q_), 140.0 (C_q_), 143.6 (C_q_), 144.1 (C_q_), 149.1 (C_q_), 165.5 (C=O) ppm. IR (neat): *ν_max_* = 3321 (N–H), 1680 (C=O), 1573 (N–O), 1465, 1343 (N–O), 1281, 1240 (C–O) 1064 (C=C, C–N) cm^−1^. MS (ESI): *m*/*z* = 297 [M–OCH_3_]^+^, 329 [M + H]^+^, 351 [M + Na]^+^, 367 [M + K]^+^. HRMS (ESI): *m*/*z* [M + H]^+^ calcd. for C_16_H_13_N_2_O_4_S 329.05905, found 329.05903. HRMS (ESI): *m*/*z* [M + Na]^+^ calcd. for C_16_H_12_N_2_NaO_4_S 351.04100, found 351.04092.

#### 3.4.11. Methyl 3-[(2-Nitrophenyl)amino]-1-benzothiophene-2-carboxylate (**3k**)

The representative experimental procedure was applied to compound **1** (200 mg, 0.965 mmol) to yield **3k** (238 mg, 75%); *R_f_* = 0.41 (PE/EtOAc 9:1); orange needles; m.p. 209–210 °C (lit. [43] 207–209 °C). ^1^H-NMR (400 MHz, CDCl_3_): *δ* = 3.92 (s, 3H, OCH_3_), 6.84 (d, *J* = 8.5 Hz, 1H, H_Ph_), 6.91 (td, *J* = 1.0 Hz, *J* = 8.3 Hz, 1H, 6-H), 7.27–7.33 (m, 2H, H_Ph_), 7.46–7.49 (m, 2H, 5-H, H_Ph_), 7.83 (d, *J* = 8.3 Hz, 1H, 4-H), 8.21 (dd, *J* = 1.3 Hz, *J* = 8.5 Hz, 1H, 7-H), 10.29 (s, 1H, NH) ppm. ^13^C-NMR (100 MHz, CDCl_3_): *δ* = 52.7 (OCH_3_), 118.6 (C_q_), 118.9 (C_Ph_), 119.7 (C-6), 123.7 (C-4), 124.7 (C_Ph_), 124.8 (C-5), 126.6 (C-7), 128.0 (C_Ph_), 133.5 (C), 135.0 (C_Ph_), 135.9 (C_q_), 139.1 (C_q_), 139.7 (C_q_), 140.5 (C_q_), 163.8 (C=O) ppm. IR (neat): *ν_max_* = 3319 (N–H), 1678 (C=O), 1575 (N–O), 1490, 1435, 1339 (N–O), 1269, 1233 (C–O), 1150 (C=C, C–N) cm^−1^. MS (ESI): *m*/*z* = 329 [M + H]^+^, 351 [M + Na]^+^, 367 [M + K]^+^. HRMS (ESI): *m*/*z* [M + H]^+^ calcd. for C_16_H_13_N_2_O_4_S 329.05905, found 329.05894. HRMS (ESI): *m*/*z* [M + Na]^+^ calcd. for C_16_H_12_N_2_NaO_4_S 351.04100, found 351.04099. HRMS (ESI): *m*/*z* [M + K]^+^ calcd for C_16_H_12_KN_2_O_4_S 367.01494, found 367.01501. ^1^H- and ^13^C-NMR spectra are in good agreement with the literature data [43].

#### 3.4.12. Methyl 3-(Phenylamino)-1-benzothiophene-2-carboxylate (**3l**)

The representative experimental procedure was applied to compound **1** (200 mg, 0.965 mmol) to yield **3l** (145 mg, 55%); *R_f_* = 0.45 (PE/EtOAc 98:2); pale beige solid; m.p. 116–117 °C. ^1^H-NMR (400 MHz, CDCl_3_): *δ* = 3.91 (s, 3H, CH_3_), 7.06–7.15 (m, 4H, H_Ph_, 6-H), 7.25–7.30 (m, 2H, H_Ph_), 7.37 (d, *J* = 7.9 Hz, 1H, 7-H), 7.41 (d, *J* = 8.1 Hz, 1H, 5-H), 7.75 (d, *J* = 8.1 Hz, 1H, 4-H), 8.74 (s, 1H, NH) ppm. ^13^C-NMR (100 MHz, CDCl_3_): *δ* = 52.0 (OCH_3_), 106.7 (C_q_), 121.8 (2 × C_Ph_), 123.4 (C_Ph_), 123.5 (C-6), 123.6 (C-4), 125.8 (C-7), 127.8 (C-5), 129.2 (2 × C_Ph_), 132.1 (C_q_), 140.2 (C_q_), 142.4 (C_q_), 146.3 (C_q_), 165.9 (C=O) ppm. IR (neat): *ν_max_* = 3320 (N–H), 1666 (C=O), 1567, 1532, 1235 (C–O), 1061 (C=C, C–N) cm^−1^. MS (ESI): *m*/*z* = 252 [M–OCH_3_]^+^, 284 [M + H]^+^, 306 [M + Na]^+^, 322 [M + K]^+^. HRMS (ESI): *m*/*z* [M + H]^+^ calcd. for C_16_H_14_NO_2_S 284.07398, found 284.07412.

#### 3.4.13. Methyl 3-[(4-Hydroxyphenyl)amino]-1-benzothiophene-2-carboxylate (**3m**)

The representative experimental procedure was applied to compound **1** (200 mg, 0.965 mmol) to yield **3m** (73 mg, 25%) using 4-iodophenol as coupling reagent; to yield **3m** (145 mg, 50%) using 4-iodophenyl acetate as coupling reagent; *R_f_* = 0.29 (PE/EtOAc 8:2); yellow solid; m.p. 166–167 °C. ^1^H-NMR (400 MHz, CDCl_3_): *δ* = 3.92 (s, 3H, OCH_3_), 4.97 (s, 1H, OH), 6.79 (d, *J* = 8.7 Hz, 2H, 2-H_Ph_, 6-H_Ph_), 7.01–7.04 (m, 2H, 3-H_Ph_, 5-H_Ph_), 7.06–7.10 (m, 1H, H-6), 7.21 (d, *J* = 8.3 Hz, 1H, 7-H), 7.37 (td, *J* = 1.0 Hz, *J* = 8.1 Hz, 1H, 5-H), 7.72 (d, *J* = 8.1 Hz, 1H, 4-H), 8.75 (s, 1H, NH). ^13^C-NMR (100 MHz, CDCl_3_): *δ* = 51.9 (OCH_3_), 103.5 (C_q_), 116.0 (C_Ph_-2, C_Ph_-6), 123.4 (C-6), 123.4 (C-4), 125.4 (C_Ph_-3, C_Ph_-5), 125.8 (C-7), 127.8 (C-5), 131.7 (C_q_), 135.2 (C_q_), 140.5 (C_q_), 148.0 (C_q_), 153.0 (C_q_), 166.3 (C=O). IR (neat): *ν_max_* = 3453 (O–H), 3307 (N–H), 1650 (C=O), 1512, 1441, 1280, 1240, 1191, 1068 (C–O, C=C, C–N, C–O–H) cm^−1^. MS (ESI): *m*/*z* = 268 [M–OCH_3_]^+^, 300 [M + H]^+^, 322 [M + Na]^+^, 338 [M + K]^+^. HRMS (ESI): *m*/*z* [M + H]^+^ calcd. for C_16_H_14_NO_3_S 300.06889, found 300.06887. HRMS (ESI): *m*/*z* [M + Na]^+^ calcd. for C_16_H_13_NNaO_3_S 322.05084, found 322.05069.

#### 3.4.14. Methyl 3-[(4-Aminophenyl)amino]-1-benzothiophene-2-carboxylate (**3n**)

The representative experimental procedure was applied to compound **1** (200 mg, 0.965 mmol) to yield **3n** (43 mg, 15%); *R_f_* = 0.17 (PE/EtOAc 8:2); brown solid; m.p. 169–172 °C. ^1^H-NMR (400 MHz, CDCl_3_): *δ* = 3.66 (br s, 2H, NH_2_), 3.91 (s, 3H, OCH_3_), 6.66 (d, *J* = 8.5 Hz, 2H, 2-H_Ph_, 6-H_Ph_), 6.99 (d, *J* = 8.5 Hz, 2H, 3-H_Ph_, 5-H_Ph_), 7.03–7.07 (m, 1H, 6-H), 7.21 (d, *J* = 8.3 Hz, 1H, 7-H), 7.35 (td, *J* = 1.0 Hz, *J* = 8.1 Hz, 1H, 5-H), 7.70 (d, *J* = 8.1 Hz, 1H, 4-H), 8.79 (br s, 1H, NH) ppm. ^13^C-NMR (100 MHz, CDCl_3_): *δ* = 51.8 (OCH_3_), 102.1 (C_q_), 115.8 (C_Ph_-2, C_Ph_-6), 123.3 (C-4, C-6), 126.0 (C_Ph_-3, C_Ph_-5), 126.0 (C-7), 127.7 (C-5), 131.7 (C_q_), 133.1 (C_q_), 140.6 (C_q_), 144.0 (C_q_), 148.6 (C_q_), 166.3 (C=O) ppm. IR (neat): *ν_max_* = 3472 (N–H), 3375 and 3312 (NH_2_), 1630 (C=O), 1572, 1514, 1439, 1390, 1237 (C–O), 1064 (C=C, C–N) cm^−1^. MS (ESI): *m*/*z* = 267 [M–OCH_3_]^+^, 299 [M + H]^+^. HRMS (ESI): *m*/*z* [M + H]^+^ calcd. for C_16_H_15_N_2_O_2_S 299.08487, found 299.08533.

#### 3.4.15. Methyl 3-{[4-(Acetylamino)phenyl]amino}-1-benzothiophene-2-carboxylate (**3o**)

The representative experimental procedure was applied to compound **1** (200 mg, 0.965 mmol) to yield **3o** (180 mg, 55%); *R_f_* = 0.18 (PE/EtOAc 7:3); yellow solid; m.p. 240–241 °C. ^1^H-NMR (400 MHz, DMSO-*d*_6_): *δ* = 2.03 (s, 3H, COCH_3_), 3.82 (s, 3H, COOCH_3_), 7.00 (d, *J* = 8.7 Hz, 2H, 2-H_Ph_, 6-H_Ph_), 7.22–7.28 (m, 2H, 6-H, 7-H), 7.46–7.51 (m, 3H, 3-H_Ph_, 5-H_Ph_, 5-H), 7.95 (d, *J* = 8.2 Hz, 1H, 4-H), 8.72 (s, 1H, NH), 9.90 (s, 1H, N*H*COCH_3_) ppm. ^13^C-NMR (100 MHz, DMSO-*d*_6_): *δ* = 23.9 (CO*C*H_3_), 51.9 (COO*C*H_3_), 106.1 (C_q_), 119.7 (C_Ph_-3, C_Ph_-5), 121.5 (C_Ph_-2, C_Ph_-6), 123.6 (C-4), 123.9 (C-6), 124.8 (C-7), 128.0 (C-5), 132.0 (C_q_), 135.0 (C_q_), 137.4 (C_q_), 139.1 (C_q_), 145.1 (C_q_), 164.3 (C=O), 168.0 (HN-C=O) ppm. IR (neat): *ν_max_* = 3279 (N–H), 3275 (N–H), 1670 (C=O), 1538, 1241 (C–O, C=C, C–N) cm^−1^. MS (ESI): *m*/*z* = 309 [M–OCH_3_]^+^, 341 [M + H]^+^, 363 [M + Na]^+^. HRMS (ESI): *m*/*z* [M + H]^+^ calcd. for C_18_H_17_N_2_O_3_S 341.09544, found 341.09575. HRMS (ESI): *m*/*z* [M + Na]^+^ calcd. for C_18_H_16_N_2_NaO_3_S 363.07738, found 363.07767.

### 3.5. Synthesis of 5a,7-Dihidro-6H-[1]benzothieno[2,3-b][1,5]benzodiazepin-6-one *(**4**)*

Compound **1** (100 mg, 0.482 mmol) was dissolved in dry dioxane (2 mL) and treated with 2-iodoaniline (160 mg, 0.724 mmol), Cs_2_CO_3_ (315 mg, 0.965 mmol), CuI (10 mg, 0.050 mmol), and l-proline (6 mg, 0.048 mmol), under argon atmosphere. Reaction mixture was stirred under reflux for 72 h. After the mixture was cooled to room temperature and diluted with water (100 mL) and extracted with ethyl acetate (2 × 50 mL). The combined organic layer was washed with brine, dried over Na_2_SO_4_, and the solvent was evaporated under reduced pressure. The resulting residue was purified by flash column chromatography (SiO_2_, eluent: petroleum ether/ethyl acetate, 7:3, *R_f_* = 0.17) to provide compound **4** as white solid. Yield 53 mg (41%), m.p. 214–215 °C. ^1^H-NMR (400 MHz, DMSO-*d*_6_): *δ* = 6.72 (d, *J* = 7.7 Hz, 1H, H_BT_), 6.95–6.99 (m, 1H, H_BT_), 7.06 (d, *J* = 7.7 Hz, 1H, H_BT_), 7.09–7.13 (m, 1H, H_BT_), 7.38–7.49 (m, 3H, H_Ar_), 8.08 (s, 1H, CH), 8.11 (d, *J* = 8.0 Hz, 1H, H_Ar_), 11.23 (br s, 1H, NH) ppm. ^13^C-NMR (100 MHz, DMSO-*d*_6_): *δ* = 108.3 (C_BT_), 109.2 (C_BT_), 121.0 (C_BT_), 121.6 (C_Ar_), 121.8 (C_BT_), 123.5 (C_Ar_), 124.6 (C_Ar_), 125.1 (C_Ar_), 125.3 (C_Ar_), 126.6 (C_q_), 128.6 (C_q_), 130.8 (C_q_), 134.5 (C_q_), 138.1 (C_q_), 153.3 (C=O) ppm. IR (neat): *ν_max_* = 3126 (N–H), 3072, 1686 (C=O), 1480, 1389, 1289, 752 (C=C, C–N, C=N) cm^−1^. MS (ESI): *m*/*z* = 267 [M + H]^+^, 389 [M + Na]^+^, 305 [M + K]^+^. HRMS (ESI): *m*/*z* [M + H]^+^ calcd. for C_15_H_11_N_2_OS 267.05866, found 267.05844. HRMS (ESI): *m*/*z* [M + Na]^+^ calcd. for C_15_H_10_N_2_NaOS 289.04060, found 289.04039. HRMS (ESI): *m*/*z* [M + K]^+^ calcd. for C_15_H_10_KN_2_OS 305.01454, found 305.01432.

### 3.6. Synthesis of Dimethyl 3,3′-Iminobis(benzo[b]thiophene-2-carboxylate) *(**5**)*

Compound **1** (100 mg, 0.482 mmol) was dissolved in dry dioxane (2 mL) and treated with compound **2** (230 mg, 0.724 mmol), Cs_2_CO_3_ (315 mg, 0.965 mmol), CuI (10 mg, 0.050 mmol), and l-proline (6 mg, 0.048 mmol) under argon atmosphere. Reaction mixture was stirred under reflux for 24 h. After the mixture was cooled to room temperature, it was diluted with water (100 mL) and extracted with ethyl acetate (2 × 50 mL). The combined organic layer was washed with brine, dried over Na_2_SO_4_, and the solvent was evaporated under reduced pressure. The resulting residue was purified by flash column chromatography (SiO_2_, eluent: petroleum ether/ethyl acetate, 95:5, *R_f_* = 0.26) to provide compound **5** as yellow solid. Yield 32 mg (25%), m.p. 236–237 °C. ^1^H-NMR (400 MHz, CDCl_3_): *δ* = 3.90 (s, 6H, 2 × OCH_3_), 7.00–7.04 (m, 2H, 2 × 6-H), 7.16 (d, *J* = 8.3 Hz, 2H, 2 × 7-H), 7.34–7.38 (m, 2H, 2 × 5-H), 7.77 (d, *J* = 8.2 Hz, 2H, 2 × 4-H), 9.84 (s, 1H, NH) ppm. ^13^C-NMR (100 MHz, CDCl_3_): *δ* = 52.3 (2 × OCH_3_), 110.6 (2 × C_q_), 123.3 (2 × C-4), 124.2 (2 × C-6), 125.0 (2 × C-7), 127.8 (2 × C-5), 133.5 (2 × C_q_), 139.5 (2 × C_q_), 142.5 (2 × C_q_), 164.7 (2 × C=O) ppm. IR (neat): *ν_max_* = 3323 (N–H), 1677 (C=O), 1574, 1438, 1277, 1228 (C–O), 1166, 1065 (C=C, C–N) cm^−1^. MS (ESI): *m*/*z* = 398 [M + H]^+^, 420 [M + Na]^+^. HRMS (ESI): *m*/*z* [M + H]^+^ calcd. for C_20_H_16_NO_4_S_2_ 398.05153, found 398.05196.

## 4. Conclusions

In summary, we demonstrated that the methyl 3-amino-1-benzothiophene-2-carboxylate **1** is a suitable precursor for modern Ullmann-type cross-coupling reactions. We successfully developed an efficient and mild CuI-catalyzed *N*-arylation reaction of functionalized aminobenzo[*b*]thiophene with a broad selection of aromatic iodides using l-proline as N,O-donor ligand, providing the coupling products **3a**–**o** in moderate to good yields. A tetracyclic diazepinone **4** was obtained from the coupling of methyl 3-amino-1-benzothiophene-2-carboxylate **1** with 2-iodoaniline in a one-pot via C–N coupling followed by immediate intramolecular cyclization. The structures of all synthesized compounds were confirmed by detailed NMR spectroscopy and HRMS investigations.

It is noteworthy that l-proline is significantly inexpensive and readily available as a promoter, and it can be easily removed from the crude reaction mixtures by simply washing them with water.

## Data Availability

The data presented in this study are available on request from the corresponding authors.

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
