# Peer review of "Mild Copper-Catalyzed, l-Proline-Promoted Cross-Coupling of Methyl 3-Amino-1-benzothiophene-2-carboxylate"

_molecules, 2021, doi:10.3390/molecules26226822_

Round 1

Reviewer 1 Report

The authors describe a Cu-catalyzed cross-coupling reaction of methyl 3-amino-1-benzothiophene-2-carboxylate with various aryl iodides. A mild reaction condition was developed to utilize a catalyst derived from CuI and L-proline to allow a smooth synthesis of N-Ar derivatives of 3-amino-1-benzothiophene -2-carboxylate. This reviewer finds out “improvement” of the content of this manuscript is required for a better understanding of the chemistry, hence, recommends a “reconsideration after major revision”.

  • Editing assistance from a native English speaker is advised before resubmitting the manuscript.
  • Please provide the compounds, starting materials and products, with suitable numbering to allow a clearing reading.
  • “Reducing the amounts of copper catalyst and L-proline as a promoter to 10 mol % gave a similar yield (Table 2, entry 6), while 5 mol % afforded slightly worse results (Table 2, entry 9).” This should be entry 7
  • The description, “To improve the reaction yield,……..heating for 24 h.”, sets in the middle of page 5 is confusing to this reviewer.
  • Table 3 and Table 4, “Substituent (R)” should be Ar.
  • The paragraph that comes after Figure 2 is also confusing to the readers.
  • On page 7, the description of the second paragraph “unprotected and protected 2-iodoanilines and 2-iodophenol were highly unexpected and provided just trace of the desired products (Table 4, entries 11–13).” is also confusing to the readers.
  • On page 7, in the third paragraph: Scheme 3 should be Scheme 4.
  • On page 7, in the third paragraph: what does “the former compound 4” mean?
  • In supplemental information, please provide NMR analytical data from HSQC, HMBC, and COSY experiments in the structural assignment of compound 4.

Reviewer 2 Report

The submitted manuscript report on “Mild Copper-Catalyzed, L-Proline-Promoted Cross-Coupling of Methyl 3-Amino-1-benzothiophene-2-carboxylate.” In this manuscript, the author has used the copper catalyst to make a C-N bond with benzothiophene derivatives. The paper is well written and all the precedented literature is cited.

I enthusiastically recommend the publication of this manuscript in Molecules, but after some minor revisions below to improve the manuscript.

  • Under Table 3, please change [d] Acetonitrile; Δ; 24 h to [e] Acetonitrile; Δ; 24 h.
  • What is the role of the ester group in benzothiophene?
  • Is any other functional group tolerated in benzothiophene?
  • Why electron-donating group in aryl halide gave a better yield than an electron-withdrawing group?
  • Does heteroaryl halide work with your current condition?

Reviewer 3 Report

This manuscript deals with the preparation of different variety of heterocyclic frameworks such as cross-coupling partners of aminobenzo[b]thiophene. To construct these substituted heterocycles, the authors utilized aminobenzo[b]thiophenecarboxylate and aryl iodides with the aid of an efficient catalytic system such as CuI and L-proline used as a ligand. In my opinion the described methodology could be an interesting extension of previous results by other groups. Since these heterocyclic cores might be biologically worthwhile and would appeal to the broad readership of “Molecules”. Though, this reviewer believes that this manuscript might be suitable for publishing after authors address few concerns which are added below.

Comments, suggestions, and questions to authors:

  1. As per my suggestion, the authors should include a separate figure with better comparison of previous result of other groups and present work.
  2. I can see that the authors used with carboxylated system, did the authors try without carboxylate in the benzothiophene core?? Why authors chose carboxylated compound??
  3. In figure 1: compound I, is it benzothiophene??
  4. I think authors should change the word to new instead of novel in the abstract.
  5. In page no. 3: Authors should include caption for Scheme 1 and some confusion in reagents and conditions like (a)…. ; (2)… (3)…. Please check and correct it as better.
  6. In page no. 3: in optimization table authors must include THF, CH3CN, and benzene.
  7. In page 5, Scheme 3: I think author should keep R-I instead of Ar-I, since substituent “R” as used for all aryl substituents in tables.
  8. Authors should check 38 there is an incomplete title.
  9. In page 6, for table 3: In conditions [d] repeated twice and it should be [e] and author must provide numbering for compound 1 in all cases.
  10. In page 6: Some confusion in table 4, the authors mentioned [a] in the caption and [b] in the yields, is it correct??
  11. In page 6, table 4: Is it entry 7 and 8 produce the same product 3m (25-50%)?? Authors should comment on this entry.
  12. In page 7, figure 2: I can see that all are aromatic systems and I suggest please include some heterocycles and compare their reactivity.
  13. In page 8, Scheme 4: I strongly encourage authors to include mechanism for benzodiazepinone ring formation with explanation for the compound 4. Also include data for intermediate such as coupling product. (in bracket instead simple structure should keep 1)
  14. I strongly suggest authors should include HSQC, HMBC, and COSY files for compound 4 in SI file, since as they mentioned in the manuscript.
  15. I suggest authors should include the 19F-NMR data as well as spectra for all the compounds where fluorine as a substituent.
  16. In page 7, 3rd paragraph authors mentioned Scheme 3 but I did not see the tetracyclic derivative 4 in scheme 3.
  17. In SI file should include DEPT135 spectra for compounds 5, 4, 3o, 3h, and
  18. Is there any reason for poor yield of compound 4 and 5??
  19. In the experimental section, authors should provide weight and % yields of the products in all cases.
  20. In supporting file (pdf), the visibility of the NMR peak values should be good and provide word file with good resolution of the spectra (check with attached pdf file) include with NMR chemical shift ranges should be 0-10 ppm (1H), 0-200 ppm (13C NMR).  

Round 2

Reviewer 1 Report

This manuscript has been amended by the authors accordingly, therefore, this reviewer recommends it be accepted for publication.

Author Response

Response to Reviewer 1 Comments

The manuscript was edited by the MDPI English editing service on 30 October 2021 (please find the certificate below).

We would like to thank the reviewer for the comments, which allowed us to improve the quality of the manuscript.

Reviewer 3 Report

The authors are addressed given concerns reasonably in the provided manuscript.

Author Response

Response to Reviewer 3 Comments

The manuscript was edited by the MDPI English editing service on 30 October 2021 (please find the certificate below).

We would like to thank the reviewer for the comments, which allowed us to improve the quality of the manuscript.
